# Complementary Nutritional Improvements of Cereal-Based Products to Reduce Postprandial Glycemic Response

**DOI:** 10.3390/nu15204401

**Published:** 2023-10-17

**Authors:** Agnès Demangeat, Hugo Hornero-Ramirez, Alexandra Meynier, Philippe Sanoner, Fiona S. Atkinson, Julie-Anne Nazare, Sophie Vinoy

**Affiliations:** 1Nutrition Research, Paris-Saclay Tech Center, Mondelez International R&D, 91400 Saclay, France; agnes.demangeat@mdlz.com (A.D.); alexandra.meynier@mdlz.com (A.M.); 2Centre de Recherche En Nutrition Humaine Rhône-Alpes, Univ-Lyon, CarMeN Laboratory, Hospices Civils de Lyon, Cens, Université Claude Bernard Lyon1, 69310 Lyon, France; ext-hugo.hornero-ramirez@chu-lyon.fr (H.H.-R.);; 3Symrise-Diana Food SAS, Campus 2, 7 Allée Ermengarde d’Anjou, ZAC Atalante Champeaux, 35011 Rennes, France; philippe.sanoner@symrise.com; 4School of Life and Environmental Sciences and the Charles Perkins Centre, The University of Sydney, Sydney, NSW 2006, Australia; fiona.atkinson@sydney.edu.au

**Keywords:** glycemic index, glycemic response, glucose homeostasis, food process, polyphenols, starch, carbohydrate

## Abstract

High glycemic response (GR) is part of cardiometabolic risk factors. Dietary polyphenols, starch digestibility, and dietary fibers could play a role in modulating GR. We formulated cereal products with high dietary fibers, polyphenols, and slowly digestible starch (SDS) contents to test their impact on the glycemic index (GI) and insulin index (II). Twelve healthy subjects were randomized in a crossover-controlled study to measure the GI and II of four biscuits according to ISO-26642(2010). Two types of biscuits were enriched with dietary fibers and polyphenols and high in SDS, and two similar control biscuits with low levels of these compounds were compared. The subjects consumed 50 g of available carbohydrates from the biscuits or from a glucose solution (reference). Glycemic and insulinemic responses were monitored for 2 h after the start of the consumption. The two enriched biscuits led to low GI and II (GI: 46 ± 5 SEM and 43 ± 4 SEM and II: 54 ± 5 SEM and 45 ± 3 SEM) when controls had moderate GI and II (GI: 57 ± 5 SEM and 58 ± 5 SEM and II: 61 ± 4 SEM and 61 ± 4 SEM). A significant difference of 11 and 15 units between the GI of enriched and control products was obtained. These differences may be explained by the polyphenol contents and high SDS levels in enriched products as well as potentially the dietary fiber content. This study provides new proposals of food formulations to induce beneficial health effects which need to be confirmed in a longer-term study in the context of the SINFONI consortium.

## 1. Introduction

Postprandial glycemic excursions, associated with high levels of insulin and lipemia, have been implicated in the etiology of non-communicable diseases such as type 2 diabetes (T2D) [1,2]. Indeed, diets that elicit lower postprandial glycemic responses (GR) and insulinemic responses (IR) are associated with several health interests, including improved insulin secretion and sensitivity, and thus enhanced glucose homeostasis [1,3,4]. Specifically, diets with a high glycemic index (GI) and/or glycemic load are associated with an increased risk of T2D and cardiovascular events in adulthood [5,6,7].

For several decades, processed foods have represented a significant and growing part of our daily diets [8]. Food processing methods are attracting increasing attention, because the processing of food products can modify the physiological fate of their components. Indeed, starch digestibility profile can be dramatically modified by controlling food processing and can improve the metabolic fate of these foods [9]. In this context, improving the nutritional properties of processed foods, as well as improving the knowledge on the precise physiological impact of food processing/ingredients on biological parameters are among the main challenges in the near future to improve diets.

For decades, dietary fibers have been recognized as one of the important dietary compounds with a health interest. Consuming dietary fiber may help prevent cardiovascular disease, T2D, and obesity [2,10,11,12,13,14]. Indeed, consuming certain types of dietary fibers has been shown to improve insulin sensitivity, as well as lipid profile [10]. Many mechanisms can interfere with these beneficial health effects [14]. Specifically, some fibers can be involved in the decrease of GI by specific mechanisms [15]. Some specific fibers, mainly viscous soluble fibers, play a significant role in managing GI and insulin secretion via the formation of viscous gels. Thereby, this leads to the slowdown of gastric emptying and the decrease in the rate of subsequent glucose absorption. Soluble fibers may also alter intestinal motility, decrease the rate of starch digestion, and reduce accessibility to α-amylase [16,17].

Dietary polyphenols display a wide range of chemical structures and are found in abundant quantities in food products including fruits, vegetables, spices, chocolate, tea, coffee, wine, and cereals [18]. Studies investigating the health interest of polyphenols showed discrepant results [19]. Several early studies reported a significant inverse correlation between the phenolic content of food products and their GR or GI [20,21]. Through in vitro approaches, potential mechanisms of action of polyphenols on the GR have been investigated. Flavonoids and phenolic acids are potent inhibitors of the activity of both α-amylase and α-glucosidase [22,23,24]. The potency of these compounds depends on their chemical structure and their bioavailability [19,24,25].

Starch is a major daily source of carbohydrates and accounts for approximately 45–60% of human energy intake. Carbohydrate quality (primarily based on a low GI approach), beyond quantity, has been shown to be critical in managing the cardiometabolic risks associated with diabetes [7,26]. Starch products may play a role in preventing hyperglycemic events, particularly if these products contain a high content of slowly digestible starch (SDS). Various studies have demonstrated that high SDS content is a key factor in diminishing postprandial glycemic and insulinemic responses in healthy adults [27,28,29] and in insulin-resistant subjects [30] in acute studies. Longer-term studies investigating the effect of increasing SDS content as a promising means of modulating a low GI diet have demonstrated a reduction in postprandial blood glucose and insulinemia and an improvement in health factors. cardiovascular risk in healthy overweight adults [31] and improve glycemic variability markers in T2D [32].

In this context, and within the framework of the SINFONI Consortium (SINFONI: synergistic innovative functional food concepts to neutralize inflammation for cardiometabolic risk prevention), the interest in improving nutritional properties of cereal-based processed foods has been addressed. The main criteria for nutritional improvement were the enrichment of these food products with dietary fibers and polyphenols, and the control of the processes to maintain a high SDS content. Fat quality has also been controlled with the increase in omega-3 in the formula. The aim of this study was to design cereal products enriched with fibers, polyphenols, and high-SDS content and improved fat quality. These cereal-based products with optimized nutritional composition were tested for glycemic and insulin index and responses compared to control cereal-based products.

## 2. Materials and Methods

### 2.1. Test Products: Nutritional Composition and Starch Digestibility

Four wheat flour-based products were prepared using rotary molded biscuit technology and extrusion technology. Two sizes of biscuits were produced for both enriched and control products with large-size cookies (enriched larger cookies (ELC) or control larger cookies (CLC)) and mini biscuits (enriched mini biscuits (EMB) and control mini biscuits (CMB)). The development of two different types of biscuits, large-size cookies and mini biscuits, was designed to test the impact of different processes on starch digestibility profiles associated or not polyphenols and fiber enrichment (Table 1). The selection of these food products was done after technology feasibility and palatability evaluations. The endpoint was to provide at least two different food products with an improved nutritional profile. The control products were developed to correspond to the enriched products based on texture and taste criteria.

The two enriched biscuits were nutritionally improved with enrichments in fibers and polyphenols and with maintaining high-SDS content and improving fat quality using rapeseed oil, compared to control biscuits in which palm oil was incorporated.

The fiber and polyphenol were extracted from cranberries, and they were provided by Diana Food Company (Champlain, Quebec, Canada). The cranberry polyphenol extract contained 44.5 g equivalent gallic acid/100 g total polyphenols in which there was a majority of proanthocyanidins A2 equivalent/100 g (PACs) with 15.9 g PACs and 48.8 g of total dietary fibers/100 g. The cranberry fiber extract contained 1.8 g equivalent gallic acid/100 g total polyphenols and 61.2 g of total dietary fibers/100 g. Total polyphenols (expressed in gallic ac equivalents) were quantified by the Folin–Ciocalteu method (ISO 14502-1) employed without previous extraction [33]. Ingredients and formulated cereals products were dispersed in the Folin reaction media and clarified before spectrophotometry to assess extractible and non-extractible phenolic groups. Cranberry polyphenol extract and fiber ingredients were characterized by High Performance Liquid Chromatography (HPLC) following typical cranberry fruit polyphenols (Cf Phenol explorer database) containing 3% of hydroxybenzoic acids, 8% of hydroxycinnamic acids, 8% of anthocyanins, 63% of flavanols (catechins and PACs), and 19% of flavonols.

The study products were made by Mondelēz in the pilot plant of the Paris–Saclay Tech Center.

### 2.2. Nutritional Composition and Starch Digestibility Analyses

The nutritional composition of the cereal-based products was analyzed with the following methodologies: total dietary fibers: AOAC 2009.01; available starch: enzymatic method (as described in the French standard V18-121, 1997); sugars from DP1 to DP7: method by high-pressure ion chromatography (HPIC); fat: Randall methodology; proteins: Kjeldahl method (×6.25) moisture. In vitro, starch digestibility was assessed using the SDS method developed by Englyst et al. [34]. This method involves several steps that simulate enzymatic digestion of carbohydrates in the stomach and the small intestine and measures the release of glucose at several time points. This method made it possible to measure the amounts of different starch and sugar fractions according to their digestibility [35].

### 2.3. Human Participants and the In Vivo Study

Twelve healthy, non-smoking volunteers between 18 and 45 years old and with a body mass index (BMI) between 18.5 to 25 kg/m^2^ participated in a randomized, open trial with a cross-over design conducted at the University of Sydney, Australia. The Human Research Ethics Committee at the University of Sydney approved the study protocol (2019/475). The experimental phase was performed from June to September 2020. Participants consumed test portions of the biscuits containing 50 g of glycemic carbohydrates with 250 mL of Evian water over a 10–12-min period. The 50 g of glycemic carbohydrates were transformed in equivalent of glucose molecules by the following calculation: (Available Starch × 1.1) + (Available Disaccharides × 1.05) + Available Monosaccharides, as recommended by the International Standards [36]. Capillary blood samples were collected in a fasted state (T0) and at regular intervals following consumption (15, 30, 45, 60, 90, and 120 min) to construct glycemic and insulinemic response curves for each test product. As recommended by the International Standards the participants recruited for the GI test were healthy adults of both sexes who satisfied two additional inclusion criteria: fasting blood glucose level < 6.0 mM and 2 h blood glucose level following the ingestion of a 50-g glucose solution < 8.9 mM, respectively [36]. The mean age and BMI of the study participants were 29.3 (SD 4.4) years and 22.8 (SD 1.6) kg/m^2^, respectively. Each volunteer consumed one portion of each test product, as well as three servings of a glucose solution (reference food), on different test days separated by a minimum of 1 day of wash out. The portion sizes for each product are shown in Table 2.

The GI of a given food reflects how much its digestible carbohydrate content raises blood glucose levels. It is defined as the incremental area under the blood glucose response curve (iAUCg) after consumption of a portion of test food providing 50 g of available carbohydrates and is expressed as a percentage of the average iAUCg for the same amount of carbohydrates from a reference food (glucose) ingested by the same subject on a separate occasion (iAUCg test food/average iAUCg reference food_100). The iAUCg is the incremental area under the blood glucose response curve (calculated over a 2 h period following ingestion of the test product: 0–120 min), ignoring the area beneath the fasting concentration (as recommended in the International Standard (ISO 26642:2010). The kinetics of glycemia over 2 h were also investigated. Moreover, the insulin index (II) was calculated with the same method as GI, by measuring the extent to which a food product raised the plasma insulin concentration [37]. The kinetics of insulinemia over 2 h were also described.

### 2.4. Statistical Analysis

Descriptive statistics (mean, SD, SEM) of all measured and calculated (GI and II) variables related to each food were determined using JMP^®^ Statistics software (version 14). A repeated-measure analysis of covariance (ANCOVA), testing the product effect, was used to determine whether there were any significant differences between the mean GI and II values and between iAUC glycemia and insulinemia of the test foods. If a statistically significant product effect was found, a post hoc multiple comparisons test was performed to identify the specific significant differences. For normally distributed data, the Tukey test was used as the post hoc test for multiple comparisons. For non-normally distributed data, a rank transformation was performed, and an analysis of ranks was done.

A repeated-measure analysis of variance (ANOVA) was used to evaluate the kinetics of glycemia and the insulinemia testing product, time as fixed effects, product × time interaction, and the subject as a random effect. If a significant product × time interaction was shown, an analysis time per time was performed on each time point of the kinetic. For kinetic data, the distribution was not normal, so a rank transformation was performed and the ranks were analyzed with a Tukey test.

## 3. Results

### 3.1. Nutrition Composition and Starch Digestibility

The nutrition composition of the four cereal products (enriched and control) is provided in Table 2. Enriched products contained 7 g of proteins/portion and 18 g of fat/portion while control products contained 4 g of proteins and 13 to 16 g of fat/portion. The available starch was similar in the four products (between 24 and 28 g/portion) but both enriched products contained significantly more slowly digestible starch (11 and 13 g/portion) than either control product (4 and 5 g/portion). Therefore, SDS accounted for 33 and 40% of available starch in ELC and EMB, respectively, whereas it corresponded to 15 and 18% of available starch in CLC and CMB, respectively. The total dietary fiber content was also much higher in enriched products (18 g/portion) compared to control products (2 and 4 g/portion). Finally, the enrichment in total polyphenols allowed us to reach a final content of nearly 2.5 g equivalent gallic acid/portion in enriched products, while there was around 0.5 g of gallic acid/portion in control products. Comparing these differences, we can estimate that the addition of polyphenols and cranberry fibers resulted in an enrichment of diets with 2.2 and 2.6 g of polyphenols, respectively, for mini biscuits and large cookies versus controls.

### 3.2. Glycemic and Insulinemic Responses

The kinetics of glycemia and insulinemia for the four products are presented as mean ± SEM in Figure 1. Baseline values of glycemia and insulinemia were not different between the products. Overall, the analysis of the glycemia kinetics over 2 h showed significant product (*p* < 0.01), time (*p* < 0.001), and product × time effects (*p* < 0.0001). When evaluating individual time points, significant differences were observed at 15, 30, 45, and 120 min with ELC inducing lower glycemia at 15 min (versus CLC and CMB), 30 min (versus CMB only), and 45 min (versus CLC only). EMB led to lower glycemia at 15 min (versus CLC and CMB) and 30 min (versus CMB only). At 120 min, ELC induced higher glycemia compared to CMB only. The analyses performed on iAUC of glycemia 0–120 min indicate a significant product effect (*p* < 0.05) (Table 3). The products enriched with fibers and polyphenols led to lower glucose iAUC compared to the control products.

There was a significant product (*p* < 0.01), time (*p* < 0.0001), and product × time effect (*p* < 0.0001) for the postprandial insulin kinetics over 2 h (Figure 1). When considering the individual time points, significant differences were observed at 15, 30, 45, 90, and 120 min. EMB-induced insulinemia was lower than for CMB at 15 and 30 min and it was higher than insulinemia following CMB at 90 min. ELC-induced insulinemia was lower compared to CMB at 15 min and by CLC and CMB at 45 min. At 120 min, ELC insulinemia was higher compared to CLC and CMB. However, despite these significant differences, the iAUC of insulinemia of the four products was not significantly different (*p* = 0.077) (Table 3).

GI and II showed differences between the products. Indeed, GI values of enriched products (ELC and EMB) were lower compared to the control products (CLC and CMB; *p* < 0.05). The GI differed by 11 to 15 points. The II values of products also differed with enriched products showing lower II values than the control products (*p* < 0.01). The differences ranged from 7 to 16 points between the products.

## 4. Discussion

In this study we investigated the impact of complementary nutritional improvements (SDS, fibers, and polyphenols) of cereal-based food products on GI and II, to demonstrate how food composition and processes can improve post-prandial metabolic responses. Presently, enriched food products containing more than 33% of SDS/available starch, 18 g of fibers, and nearly 2.5 g total polyphenol (expressed in gallic acid equivalent) induced a significant decrease in acute postprandial glycemic and insulinemic responses compared to control cereal-based foods by 30% and 23%, respectively, when considering the mean iAUC of both treatments. The GI values of enriched products were significantly reduced by more than 10 points and the II values by 7 and 16 points compared to control products, with none of the four products exacerbating insulin secretion in parallel to the GR. The reduced glucose excursions associated with a non-exacerbated insulin response have been shown to contribute to the long-term beneficial health effects in non-diseased subjects [38,39]. Several features from the selected nutritional improvement of the tested cereal products may have contributed to this beneficial metabolic impact. In addition, the enriched food products contained 3 g more proteins per portion and improved the lipid profile with nearly no saturated fat and 2 g of *n*-3 polyunsaturated fat compared to control products. These additional improvements may be impactful in a longer-term effect but have not been shown to interfere with acute metabolic response [16]. According to previous studies on cereal-based products, the control products induced a medium GI due to their content of SDS and fat [40].

The content of SDS was obtained by the strict control of the food process. Indeed, it has been clearly shown for several years that food processes modify the starch structure configuration inside the food matrix depending on its level of gelatinization [16,39,41]. Several food processes used to make biscuits have been recently compared and showed that rotary molding technology which needs low water addition and smoother technological conditions is favorable to starch preservation in its native state by limiting starch gelatinization and maintaining a higher SDS content [40]. The originality of the present work is, that by using the same favorable technology (rotary molding), and by controlling some specific parts of the process, we were able to obtain a wide range of SDS, from 15 to 40% of available starch. The main differences in food processes between the enriched biscuits and CMC were, the cooking duration and the treatment of the wheat flour used in the different formulas. For the CLC, the extrusion process was used which has been recognized as a unfavorable process to preserve native starch from gelatinization [40]. In the control biscuits, extruded wheat flour was used to increase starch gelatinization and therefore reduce the SDS content. Successfully, the enriched biscuits contained the highest amount of SDS with 11 and 13 g per portion while control products contained only 4 and 5 g per portion. This range of SDS quantities has been studied in the past and showed a significant negative relationship between SDS content and postprandial GR in cereal-based foods, i.e., the highest the SDS content, the lowest the postprandial GR [28,42,43].

The total amount of polyphenols in the enriched products reached nearly 2.5 g gallic acid equivalent per portion with a high amount of proanthocyanidins and flavonoids. This diet enrichment corresponds to the daily intake of total polyphenols in France (2.1 +/− 1 g/d) assayed by the same method [18]. Polyphenols are able to slow down starch digestion by reducing the activities of α-amylase and α-glucosidase, both major digestive enzymes involved in carbohydrate digestion [44]. Polyphenol tannins containing polymer diphenol groups as proanthocyanidins can create starch–polyphenols complexes in the food matrix which may modify starch digestibility [45]. However, polyphenols with different physico-chemical structures showed great differences in inhibition of enzyme activities and in complexing capacity [46]. Some studies evaluating the groups of flavonoids and proanthocyanidins showed that specific interactions with both digestive enzymes (α-amylase and α-glucosidase) have been shown to reduce their activity through specific bonds on their binding sites or nearby them to exclude the binding of the substrate [45,47]. The response to the Folin–Ciocalteu test showed that the reactive phenolic groups were no longer extractable, but were still present in the enriched cooked products. However, traditional methods for measuring proanthocyanidins were no more effective. The characterization of the proanthocyanidin polymer units involved in these interactions is the subject of specific verifications with an adapted analytical characterization after hydrolysis in a complementary study. The main effects were observed at the small intestine level. Even if the potential mechanistic effect of polyphenols on starch digestion has been well documented in vitro, it is difficult to draw conclusions on the dose–response effect of polyphenol enrichment through inclusion in cereal-based products [45,47]. Furthermore, in vitro and mouse models, polyphenols studies have been demonstrated to potentially modulate glucose transport and absorption, particularly at the intestinal level [48]. Recently, a study showed no significant effect of a high-carbohydrate cereal food enriched with cranberry inclusions on postprandial glycemic response [49]. In this last study, the tested enrichment was 0.6 g gallic acid equivalent per portion when the level obtained in the present work is 2.5 g gallic acid equivalent for the enriched products and 0.5 gallic acid equivalent for the control foods. In Smith’s study, enriched cereals bars contained higher proportions of simple sugars, and starch–polyphenols interactions were less favored than in a cooked dough, and this also could explain the differential impact.

In the studied products, only 16% of dietary fibers were measured as soluble fibers in the enriched products, with 3 g per portion. These fibers might contribute to limiting the accessibility of starch by the digestive enzymes and increasing the viscosity of the bolus in the stomach, if parts of these soluble fibers develop viscosity in the digestive tract [50]. The addition of dietary fibers extracted from cranberries increased mainly insoluble parts of the enriched products. It is worth noting that the fiber content does not alter the glycemic load of the portion as the size of the portion was adjusted based on its available carbohydrate content. In this context, the portion of the enriched products which contained more fibers was increased to provide 50 g of available carbohydrate, as is required in GI study design.

The long-term effects of these enriched products, characterized by lower GI and II, need to be tested to evaluate the full potential of these processed products with improved nutritional profiles in the prevention of cardiometabolic risk. The lower postprandial glycemic and insulin responses may be associated with additional metabolic effects provided by selected active compounds, mainly fibers and polyphenols, which are been shown to interfere with oxidative stress and inflammation processes and could influence gut microbiota composition and activity [19,51,52,53]. Dietary fibers interfere with the digestive tract on digestion processes and can also play through influencing gut microbiota composition and activity [51,54].

## 5. Conclusions

Globally, the nutritional improvements of the enriched products led to an improvement in postprandial glycemic and insulin responses in healthy subjects. This beneficial short-term effect may be based on complementary mechanistic effects on glucose metabolism by playing on carbohydrate digestibility:

SDS, due to the limitation of its gelatinization during the cooking process, is digested slowly in the small intestine, leading to the slow release of glucose in the bloodstream.

The cranberry polyphenols included in the cereal matrix can interact with α-amylase and α-glucosidase to slow down starch digestion. As an important fraction of the available starch in the products, the rapidly digestible starch fraction is gelatinized, and the polyphenols may interfere favorably with this starch fraction to slow down its digestion.

Part of the fibers are soluble and they might interfere with the gastric bolus and potentially slightly slow down the starch digestion (excepted minimal effect).

The significant short-term effects observed have been established in healthy subjects and could also prove valuable for individuals at cardiometabolic risk or with metabolic disturbances, including individuals with diabetes. Further research is needed to fully evaluate the potential implications of these findings in a clinical context and after several weeks of ingestion and to elucidate the underlying mechanisms and potential food synergy interactions on health.

## Figures and Tables

**Figure 1 nutrients-15-04401-f001:**
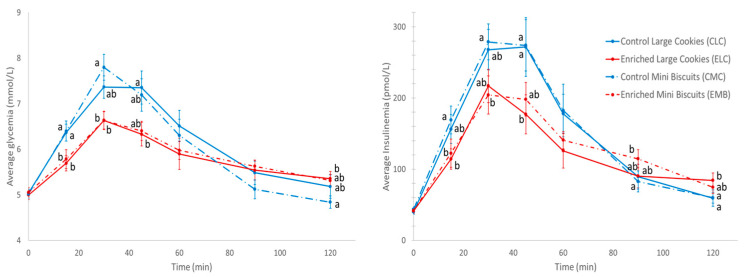
Kinetics of the postprandial glycemic response and insulinemic response after the ingestion of the enriched larger cookies (ELC), control larger cookies (CLC), and enriched mini biscuits (EMB) and control mini biscuits (CMB). *n* = 12 subjects, results are shown mean ± SEM. Different letter showed significant statistical difference.

**Table 1 nutrients-15-04401-t001:** Details of the ingredients and food processing of the four cereal products.

	Ingredients	Processing	Cooking
**Enriched Large Cookie (ELC)**	Wheat flour, sugar, cranberry crisps, rapeseed oil, cranberry fibers, dried cranberries, buckwheat, polyphenol extract from cranberries, ammonium bicarbonate, pyrophosphate, sodium bicarbonate, salt, inactive dry yeast, DATEM (diacetyl tartaric acid ester of mono-diglycerides), soy lecithin	Rotary molded (Vuurslag rotary)	Tunnel oven APV baker; baking time: 5.5 min; temperatures: fluctuating between 140 and 190 °C
**Enriched Mini Biscuit (EMB)**	Wheat flour, cranberry fibers, sugar, rapeseed oil, buckwheat, polyphenol extract from cranberries, ammonium bicarbonate, soy lecithin, pyrophosphate, sodium bicarbonate, DATEM (diacetyl tartaric acid ester of mono-diglycerides), inactive dry yeast	Rotary molded (Vuurslag rotary)	Tunnel oven APV baker; baking time: 6 min; temperatures: fluctuating between 140 and 190 °C
**Control Large Cookie (CLC)**	Extruded wheat flour, sugar, palm oil, wheat flour, dried cranberries, DATEM (diacetyl tartaric acid ester of mono-diglycerides), salt, brown coloring	Extrusion: co-extruder Rhéon KN135	Tunnel oven Imaforni; baking time: 9 min; temperatures: 200 °C
**Control Mini Biscuit (CMB)**	Extruded wheat flour, sugar, wheat flour, palm oil, DATEM (diacetyl tartaric acid ester of mono-diglycerides), salt, brown coloring	Rotary molded (Vuurslag rotary)	Tunnel oven APV baker; baking time: 6 min; temperatures: fluctuating between 140 and 190 °C

**Table 2 nutrients-15-04401-t002:** Nutrition composition and starch digestibility.

	Control Large Cookies (CLC)	Control Mini Biscuits (CMB)	Enriched Large Cookies (ELC)	Enriched Mini Biscuits (EMB)
Portion size (g)	77	67	103	99
Energy (kcal/portion)	376	328	428	426
Moisture (g/portion)	3	2	5	3
Proteins (g/portion)	4	4	7	7
Fat (g/portion)	16	12	18	18
-Saturated	8	6	1	1
-Polyunsaturated n-3	0	0	2	2
Available carbohydrates (g/portion eq. Glucose *)	50	50	50	50
Available starch (g/portion)	24	26	25	28
Sugars (g/portion)	22	18	19	17
Total dietary fibers (g/portion)	2	2	19	18
Insoluble dietary fibers (g/portion)	1	1	16	15
Soluble dietary fibers (g/portion)	1	1	3	3
SDS (g/portion eq glucose)	4	5	11	13
SDS/Av. Starch (%)	15	18	33	40
Total polyphenols (g eq gallic acid/portion)	0.2	0.4	2.6	2.2

* g/portion eq. Glucose: equivalent of glucose molecules is calculated by (Available Starch × 1.1) + (Available Disaccharides × 1.05) + Available Monosaccharides.

**Table 3 nutrients-15-04401-t003:** Participants’ postprandial metabolic response parameters (*n* = 12).

Blood Glucose and Insulin Parameters	CLC	CMB	ELC	EMB	*p*-Value for Comparison of Four Products	*p*-Value Comparison of Control Products versus Enriched Products
Baseline blood glucose level (mmol/L)	5.0 ± 0.1	5.0 ± 0.1	5.0 ± 0.1	5.1 ± 0.1	NS	NS
iAUC (g) (mmol × min/L)	143 ± 22 ^b^	134 ± 18 ^b^	98 ± 16 ^a^	97 ± 9 ^a^	*p* < 0.05	*p* < 0.01
GI (%)	58 ± 5 ^b^	57 ± 5 ^b^	43 ± 4 ^a^	46 ± 5 ^a^	*p* < 0.05	*p* < 0.01
Baseline insulin level (pmol/L)	40 ± 4	44 ± 4	41 ± 4	42 ± 4	NS	NS
iAUC(ins) (pmol × min/L)	13,550 ± 1907 ^b^	13,431 ± 1713 ^b^	9807 ± 1375 ^a^	10,888 ± 1249 ^a^	NS	*p* < 0.05
II (%)	61 ± 4 ^b^	61 ± 4 ^b^	54 ± 5 ^ab^	45 ± 3 ^a^	*p* < 0.05	*p* < 0.01

CLC: control large cookies; CMB: control mini-biscuits; ELC: enriched large cookies; EMB: enriched mini-biscuit. Glucose parameters: baseline, incremental area under the curve (iAUC (g)), and glycemic index (GI). Insulin parameters: baseline, incremental area under the curve (iAUC (ins)), and insulin index (II). Statistical differences (*p* < 0.05) are represented by different letters a and b a significant letter; a and b signifying a significant difference between enriched and control cookies and ab for an intra-group difference. Values expressed as mean ± SEM.

## Data Availability

Data presented in this study are available upon request to the corresponding author. The data are not publicly available.

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
