# Peer review of "Complementary Nutritional Improvements of Cereal-Based Products to Reduce Postprandial Glycemic Response"

_nutrients, 2023, doi:10.3390/nu15204401_

Round 1
Reviewer 1 Report
The study investigated the effect of a combination of SDS, polyphenols and dietary fiber (extracted from cranberries) on glucose and insulin response for two flour-based biscuits. Glycemic, and insulin index were determined in 12 healthy young adults.
Introduction
Second paragraph, processing is a very broad area and ultra processing is not relevant here, describe processes linked to SDS.
Material and methods
Describe the differences in the manufacturing processes of the four products in detail and how this affects the amount of SDS in the product.
The purpose of the entire arrangement with two test products is unclear. Explain the difference, in addition to size, between the two test products and what you want to study. The same applies to the two control products.
Is it not recommended that the wash out period is more than a day?
Result
The nutrition composition is in Table 2, not in Table 3.
Explain the point of having control products when there are differences in protein, fat and sugar content between them and the test products.
The difference in energy between CLC and CMB is unclear.
Explain the calculation of starch eq glucose in a footnote.
Unfortunately Figure 1 was not included in the downloaded material. According to the text the glucose and insulin responses after four products are included, the glucose and insulin response after glucose should also be included as GI is calculated.
Add data on baseline, glucose iAUC(g) and iAUC(ins) in Table 3
Diskussion
The control products gave a medium high GI, please reason something about what it is due to.
Bring up the differences between the two test products that did not produce any significant differences in GI.
Author Response
The study investigated the effect of a combination of SDS, polyphenols and dietary fiber (extracted from cranberries) on glucose and insulin response for two flour-based biscuits. Glycemic, and insulin index were determined in 12 healthy young adults.
The authors thank the reviewer for these very constructive remarks and questions which helped to clarify important points of the manuscript.
Introduction
Second paragraph, processing is a very broad area and ultra processing is not relevant here, describe processes linked to SDS.
We thank the reviewer for this remark. The sentence “There is growing controversy regarding the potential health risks of certain food processing technologies”, as well as the reference to Hall et al. 2019 paper, have been removed.
The following sentence has been added to focus on starch digestibility, as recommended: “Indeed, starch digestibility profile can be dramatically modified by controlling food processing and can improve the metabolic fate of these foods”. The reference of Singh et al. 2003 has been added.
Material and methods
Describe the differences in the manufacturing processes of the four products in detail and how this affects the amount of SDS in the product.
The details of the food processes are summarized in the table 1. The authors did not mention the details of the information provided in the table to avoid duplication of the information, as it is requested by the editor.
The purpose of the entire arrangement with two test products is unclear. Explain the difference, in addition to size, between the two test products and what you want to study. The same applies to the two control products.
We thank the reviewer for this point which was missing in the submitted version of the manuscript. The following sentences have been added.
“The development of two different types of biscuits, large size cookies and mini-biscuits was designed to test the impact of different processes on starch digestibility profiles associated or not to polyphenols and fiber enrichment. The selection of these food products was done after technology feasibility and palatability evaluations. The end point was to provide at least two different food products with an improved nutritional profile. The control products were developed to correspond to the enriched products based on texture and taste criteria.”
Is it not recommended that the wash out period is more than a day?
The wash-out period is very limited as the level of intervention is very limited. The subject is requested to eat a portion of food product at a fasting stage at an equivalent of breakfast time and the blood sampling is very limited because it is only capillary blood samples collected 7 times. At each time, only few drops of blood are taken from a finger prick. In addition, the protocol has been validated by the ethical committee.
Result
The nutrition composition is in Table 2, not in Table 3.
Thank you for your vigilance.
Explain the point of having control products when there are differences in protein, fat and sugar content between them and the test products.
The additional nutritional differences between control and enriched food products are much less important, especially in acute metabolic response. However, tha nks to your comment, we added this sentence in the discussion.
“In addition, the enriched food products contained 3g more proteins per portion and improved lipid profile with nearly no saturated fat and 2g of n-3 polyunsaturated fat compared to control products. These additional improvements may be impactful in a longer-term effect but has not been shown to interfere with acute metabolic response [16].”
The difference in energy between CLC and CMB is unclear.
The energy content of the control products is lower than the enriched products mainly because of the difference in fat and fibers contents which provide 9kcal/g and 2 kcal/g, respectively.
Explain the calculation of starch eq glucose in a footnote.
Thank you to mention this point which is important for the design of the study. We added this sentence in the material and methods:
“The 50 g of glycemic carbohydrates were transformed in equivalent of glucose molecules by the following calculation: (Available Starch x 1.1) + (Available Disaccharides x 1.05) + Available Monosaccharides, as recommended by the International Standards (ISO 26642:2010).”
Unfortunately Figure 1 was not included in the downloaded material. According to the text the glucose and insulin responses after four products are included, the glucose and insulin response after glucose should also be included as GI is calculated.
Thank you for your vigilance. There was a problem during the edition of published format, as the figure 1 was transmitted to the editor is separate file, as required. The graphs are now added in the manuscript.
Regarding the point of adding the glucose solution in the graphs. We did not do it because glucose solution was used for GI calculation only and is the same for the 4 products. In fact, the main objective of the study is the comparison of the acute metabolic response of the 4 cereal-based products, and especially enriched compared to control products. For your information, please find enclosed the graphs with the glucose solution.
Add data on baseline, glucose iAUC(g) and iAUC(ins) in Table 3
The present study has been performed in acute conditions. Therefore, the iAUC at baseline is equal to zero, as the AUC of baseline is subtracted from the incremental AUC of glucose and insulin. The following sentence is written in the protocol description. Please let us know if we need to clarify this point. “the iAUCg is the incremental area under the blood glucose response curve (calculated over a 2-h period following ingestion of the test product: 0–120 min), ignoring the area beneath the fasting concentration (as recommended in the International Standard (ISO 26642:2010).”
We used the same principle for the calculation of the iAUC insulin.
Discussion
The control products gave a medium high GI, please reason something about what it is due to.
The GI of the control products are in line with previous works published. To clarify this important point, we added this following sentence:
”According to previous studies on cereal-based products, the control products induced a medium GI due to their content in SDS and fat [41]. “
Bring up the differences between the two test products that did not produce any significant differences in GI.
The additional nutritional differences between control and enriched food products are much less important, especially in acute metabolic response. However, tha nks to your comment, we added this sentence in the discussion.
“In addition, the enriched food products contained 3g more proteins per portion and improved lipid profile with nearly no saturated fat and 2g of n-3 polyunsaturated fat compared to control products. These additional improvements may be impactful in a longer term effect but has not been shown to interfere with acute metabolic response [16].”

Reviewer 2 Report
The present study is well structured and detailed. It appears that a control was done with the same available carb content (50g) so slightly different portion sizes were used for the control and enriched cookies.
As a limitation I could state the small sample size (12 healthy subjects, aged 18-45 years, normal BMI, from Sydney Australia) and for this reason a generalization of the results cannot be made.
In the study 4 different products have been used: 2 enriched cookies (1 large and 1 small) and 2 control cookies (1 large and 1 small). In the enriched cookies, rapeseed oil was used as source of fat to enrich them in Ω3 unsaturated fats, a point that was not studied and was not taken into account in the study which mainly aimed to calculate the glycemic load and the glycemic response in proportion to the content of the product in polyphenols and dietary fibers. In the same time in the control cookies, palm oil, a source of ω6 and saturated fat, was used as a fat source. The different types of fats were added in other quantities and indeed not in proportion to their size.
It would be useful to include a graph with a graphic representation of the glycemic load or glucose response depending on each product could be included.
In total it is an interesting study and it contributes to the general effort to produce food products suitable for people on diabetes or not but mainly suitable for a lower postprandial glycemic response.
Author Response
The present study is well structured and detailed. It appears that a control was done with the same available carb content (50g) so slightly different portion sizes were used for the control and enriched cookies.
We thank the reviewer for these very kind comments and remarks.
As a limitation I could state the small sample size (12 healthy subjects, aged 18-45 years, normal BMI, from Sydney Australia) and for this reason a generalization of the results cannot be made.
Thank you for this important comment. The interest of this work would be to test these products on more subjects in a longer-term period. In addition, the authors wanted to clarify that the number of subjects in the GI determination has been defined in the international guidelines published in 2010, where a minimum number of 10 subjects is required for GI. We included 12 subjects in case we would lose some subjects.
In the study 4 different products have been used: 2 enriched cookies (1 large and 1 small) and 2 control cookies (1 large and 1 small). In the enriched cookies, rapeseed oil was used as source of fat to enrich them in Ω3 unsaturated fats, a point that was not studied and was not taken into account in the study which mainly aimed to calculate the glycemic load and the glycemic response in proportion to the content of the product in polyphenols and dietary fibers. In the same time in the control cookies, palm oil, a source of ω6 and saturated fat, was used as a fat source. The different types of fats were added in other quantities and indeed not in proportion to their size.
Thank you for this remark. The fat quantity is not exactly the same in control and enriched food products due to the differences in processes and to reach an acceptable palatability for enriched food products. The authors did not comment much this aspect of quantity and quality of fat because these levels are not different enough from the control products to interfere with the acute glycemic response. For reference, you may find additional information in Meynier et al. 2015. In a long-term approach, these fat differences may be interesting to be investigated.
It would be useful to include a graph with a graphic representation of the glycemic load or glucose response depending on each product could be included.
Thank you for your remark. The graphs of glycemic and insulin responses have been added to the manuscript.
In total it is an interesting study and it contributes to the general effort to produce food products suitable for people on diabetes or not but mainly suitable for a lower postprandial glycemic response.
